# Linking belief in a just world and psychological capital to psychological basic needs satisfaction and mental health of young adults living with HIV: A comparative analysis

**Martin Mabunda Baluku**[1]*, **Samuel Ouma**[2], **Brian Iredale**[3], **Gerald Mukisa Nsereko**[1], **Joanita Nangendo**[4], **Stuart Kwikiriza**[3], **John Kiweewa**[5]

1 Department of Educational Social, and Organizational Psychology, School of Psychology, Makerere University, Kampala, Uganda, 2 Department of Mental Health and Community Psychology, School of Psychology, Makerere University, Kampala, Uganda, 3 Nurture Africa Medical Center, Kampala, Uganda, 4 Mak-BSSR Program, School of Medicine, Makerere University, Kampala, Uganda, 5 School of Education and Human Development, Fairfield University, Fairfield, CT, United States of America

* mbaluku1@gmail.com, martin.baluku@mak.c.ug

**Data Availability Statement:** The data has been uploaded as a supplementary file.

## Abstract

People living with HIV (PWH) have higher risks for negative experiences and emotions such as discrimination, self-blame, and denial, which make them vulnerable to mental health problems. Those living with HIV and are in the developmental stage of young adulthood (18–39 years) have added psychological challenges arising from the pressure to perform the developmental tasks of this stage, which may gratify or thwart basic psychological needs (BPNs) and impact their mental health. The study examined whether positive psychological attributes, including a belief in a just world (BJW) and psychological capital, could be resources for satisfying basic psychological needs, hence boosting the mental health of Young Adults Living with HIV (YALWH). A comparison sample of university students was also included in the study. The results show that BJW was positively directly associated with BPN satisfaction, BPN frustration, and mental health (flourishing aspect) in the student sample but not in the YALWH sample. Psychological capital was appositively associated with BPN satisfaction and flourishing in the student sample. On the other hand, psychological capital is only positively associated with BPN satisfaction and negatively with BPN frustration and distress in the YALWH sample. The serial medial analysis revealed that the effects of BJW on mental health (flourishing) are mediated by psychological capital and BPN satisfaction in both samples. On the other hand, the effects of BJW on distress are mediated by psychological capital and BPN frustration, again in both samples. Incorporating interventions for strengthening positive psychological attributes could be helpful for YALWH and other young adults to attain desirable developmental outcomes for this stage and their mental health.

**Funding:** The authors disclosed receipt of the following financial support for the research, authorship, and/or publication of this article: Research reported in this publication was supported by the Fogarty International Center (FIC), the National Institute of Alcohol Abuse and Alcoholism (NIAAA), and the National Institute of Mental Health (NIMH), of the National Institutes of Health (NIH) under the Award Number D43 TW011304. The content is solely the responsibility of the authors and does not necessarily represent the official views of the National Institutes of Health (https://www.nih.gov/).

**Competing interests:** The authors have declared that no competing interests exist.

## Introduction

The commitment to "ending inequalities and getting on track to end AIDS by 2030" calls for an integrated HIV response [1]. However, the population of Young Adults Living with HIV (YALWH) is at risk of being forgotten or not receiving a range of services that could be essential at their development stage. This population is already behind most other age groups concerning antiretroviral therapy (ART) coverage [2]. By the end of 2022, an estimated 0.7% of the world population of people aged 15–49 years were living with HIV [3]. In Uganda, adolescents and young adults constitute about 12% of the people living with HIV. Yet, they tend to have poor outcomes on the HIV care continuum partly due to the challenges of transitioning to adult HIV care [4].

From a developmental psychology perspective, young adulthood includes emerging adulthood "18–25 years" and early adulthood "26–39 years)" [5–7]. It follows the adolescent stage when individuals go through emotional and behavioral upheaval [8]; hence, young adulthood becomes the stage for moving into stable equilibrium as one navigates new environments and personal growth tasks [9]. In line with the Self-Determination Theory (SDT) of Basic Psychological Needs "BPNs" [10–12], some developmental tasks of young adulthood, including achieving autonomy, establishing a career, and finding intimacy, correspond to the BPNs. The BPNs include the need for autonomy, the need for relatedness, and the need for competence. These psychological needs are considered drivers for motivation and psychological health [13, 14]. Mehta and colleagues described early adulthood as entailing some of the most intense and demanding but equally rewarding experiences [15]. Positive experiences of achievement and satisfaction of needs yield happiness and success in later tasks, while failure to achieve developmental tasks and gratify the corresponding needs yield unhappiness, disapproval by society, and difficulty in later stages of life [16], signaling developmental crises.

The psychosocial and developmental challenges experienced by YALWH have implications for psychological need satisfaction and psychological growth. Concerning the need for autonomy, available evidence suggests that autonomy support is essential in reducing HIV stigma [17]. Regarding the need for relatedness, the literature suggests that young PWH experience delayed sexual maturation [18] while also challenged by the desire to have intimate relationships while at the same time maintaining the confidentiality of their HIV status [19, 20]. Regarding the need for competence and career development, the schooling and work environments tend to be stigmatizing to PWH [19, 21] in addition to the lack of privacy and supportive relationships in the school or work settings [4, 22]. The majority of PWH are of working age [23] but are often left out on employment opportunities through discrimination and social exclusion tendencies [24], hence high unemployment levels and engagement in precarious and hazardous employment [25]. This situation not only affects the satisfaction of the need for competence, but it also impacts HIV care outcomes, including health, HIV care retention, and medication adherence [23, 24]. Despite their usefulness to health, psychological wellness, and personal growth, there is surprisingly limited research on the satisfaction of BPNs among YALWH.

Based on positive psychology literature, the present study examines how positive psychological attributes, including a Belief in a Just World (BJW) and psychological capital influence the satisfaction of BPNs among YALWH. Positive psychology relates to conditions and processes that lead to flourishing and optimal functioning [26, 27]. Therefore, positive psychological attributes are psychological characteristics of individuals that enable them to flourish and attain desirable outcomes. BJW is the belief that the world is fair where people get what they deserve [28, 29]. This belief boosts trust, hope, and confidence in people's future [30], enabling individuals to confront issues in their environment as if it is orderly and stable [29]. In this

sense, BJW is considered a personal resource that can help individuals cope with uncertainty in life and work [31].

Similarly, psychological capital comprises psychological resources that define an individual's mental strength [32, 33]. The resources that constitute psychological capital include self-efficacy, hope, optimism, and resilience, which tend to have a more significant impact when considered together [32, 33]. Given that the common thread in positive psychological resources is the ability to overcome distress and achieve psychological growth, the current study examines whether these constructs could be essential for the satisfaction of BPNs and the mental health of YALWH. The study compares YALWH with a sample of students to generate preliminary evidence of whether the role of BJW and psychological capital in BPN satisfaction and mental health in the YALWH differs from other populations.

## Theory and hypotheses

Attaining behavioral outcomes is a function of personal agency and environmental influences [34, 35]. The present study focuses on the individual agency that might stem from a belief in a just world (believing that the world is just and fair) and psychological capital (one's mental strength). Therefore, the study is based on the Self-Determination Theory "SDT" [36, 37], which offers important insights into what motivates and sustains human behavior. Accordingly, human actions are influenced by attitudes and goals that vary in nature. Some aspirations are internally generated by inherent interest, i.e., "intrinsic motivation," while others are elicited by envisaged separable outcomes, i.e., "extrinsic motivation" [38, 39]. SDT presents self-motivation or autonomous motivation [12, 40] as the highest form of motivation that drives active engagement, involvement, and persistence in activities. This form of motivation is essential for psychological development as one pursues the satisfaction of BPNs [41, 42]. Autonomous motivation comprises both intrinsic and some forms of extrinsic motivation characteristics whereby individuals identify with the value of the behavior and integrate it into the sense of self [12]. In such situations, the motivation for behavior is self-determined and is only enhanced or undermined by social and environmental factors.

The SDT further posits that the intrinsic inspiration for engaging in a given behavior is driven by the aspiration to satisfy three Basic Psychological Needs (BPNs): autonomy, relatedness, and competence [39, 41]. Further, building on the positive psychology theory [32, 43, 44], the present study seeks to test whether positive resources nested in BJW and psychological capital enhance the satisfaction of BPNs. The qualities in these positive constructs are good for understanding and relieving human suffering and can nourish good lives [44]. The gratification of psychological needs is related to flourishing, as reflected in motivation and wellbeing [14, 45]. Consequently, the positive psychological attributes may have direct and indirect effects (through the gratification of BPNs) on mental health and other HIV-care outcomes.

## The role of belief in a just world

Belief in a Just World (BJW) is a basic trust that the world is a just place where people's rewards, punishments, and other outcomes fit what they deserve [29]. This fundamental belief makes people perceive that their environment is stable and orderly. Hence, individuals tend to think they can be treated justly and fairly by others [46], which facilitates the development of a sense of trust, hope, and confidence in the future [30]. The qualities enshrined in BJW enable individuals to cope with the uncertainty of life and bounce back from adverse situations [31, 47]. In addition, a strong BJW enables individuals to have a high level of perceived control over their lives [31, 48] and a positive perception of the future [48]. Moreover, individuals with a strong BJW are more likely to feel better about themselves [49] and have fewer perceptions

of risk related to their careers [50, 51]. Thus, young adults living with HIV would perceive fewer injustices in the form of stigma and discrimination.

The belief in one's own justice (personal BJW) needs to be distinguished from the belief in justice in general (general BJW) since personal BJW and general BJW have different and, sometimes, opposite effects [52]. Specifically, only personal BJW tends to have a more consistent impact on an individual's level of perceived victimization and wellbeing, as well as on other positive life and work outcomes [50, 53–55] including in situations of disadvantage, such as during natural disasters, in unjust situations such as bullying and violence, and in uncertain career situations [47, 52]. In addition, the vital adaptive functions of personal BJW [31, 46] tend to buffer against adverse psychological outcomes [47].

This study hypothesizes that young adults living with HIV can be resilient to psycho-developmental challenges and gratify their BPNs. Overall, BJW is associated with perceptions of trust, optimism, gratitude, and meaningful interpretation and events, which promotes mental health [53, 56]. In undesirable or unjust situations, a strong BJW leads to positive reactions. For example, in their study, Liu and colleagues found that high levels of BJW mitigated the psychological effects of perceived discrimination [57]. On the other hand, low levels of BJW can result in negative reactions and outcomes. For example, a decline in BJW among PWH is associated with reduced perceived control over one's own life, which may have negative consequences for motivation to seek growth, health, and related outcomes [58]. Concerning the need for competence and career development Dzuka and Dalbert [59] found that high personal BJW motivated unemployed young people to downplay unfairness in their unemployment situation [59]. Such positive reactions to otherwise stressful situations are possible because BJW enables individuals to interpret events meaningfully [54]. Even in times of change or crisis, BJW allows individuals to continue exhibiting good behavior since personal BJW may represent the belief that positive behavior can protect them from adverse outcomes [60], hence the motivation to exert effort towards satisfying the BPNs.

Satisfaction of BPNs is linked to positive developmental and health outcomes. The SDT posits that satisfaction of BPNs facilitates mental health [39, 61], which has been confirmed in several empirical studies [62, 63]. A study among Chinese adolescents revealed that general and personal BJW was associated with greater satisfaction for the BPNs, which explained the variance in meaning in life [64]. This finding suggests that BPNs could serve as a mediator between BJW and mental health outcomes.

BJW also has its dark side. One possible way of restoring justice psychologically, where injustice is perceived, is by blaming the victim by downplaying the victim's character and thus deserving of the negative consequences [65]. Therefore, people who strongly believe in a just world are more likely to attribute the victim's suffering to the victim's character and actions [66, 67]. Individuals with a strong BJW are likely to blame themselves when it comes to the self as a victim. This phenomenon is mainly observed among rape victims [68, 69]. Concerning HIV, individuals are likely to blame themselves or feel guilty for their serostatus. However, blaming one's own character for the negative experiences or consequences negatively impacts coping and, consequently, may be detrimental to the satisfaction of BPNs and mental health. BJW is also associated with stigma, denial, and complacency [70, 71], thus may catalyze the development of mental health problems among PWH. Therefore, whereas BJW may be essential in the satisfaction of BPNs and is beneficial for mental health, it is also likely to have detrimental outcomes where individuals perceive their own suffering as deserved.

## The role of psychological capital

Psychological capital is rooted in positive organizational psychology [32, 33, 72] and is considered to be a positive developmental state encompassing four mental resources, including confidence (self-efficacy), hope, optimism, and resilience. These psychological resources are related and tend to move together, so an increase in one is likely to increase the others [33]. Psychological capital reflects an individual's ability to undertake and succeed in challenging tasks (representing self-efficacy). Psychological capital also represents the ability to make positive attributions and realistic expectations of success now and in the future (optimism), perseverance toward goals, and the ability to create alternative pathways to achieve goals (hope). Finally, it reflects the ability to sustain action and bounce back when facing adversity to attain success (resilience) [73, 74]. Psychological capital gives individuals a sense of control when pursuing goals [75].

The positivity from psychological capital facilitates attaining extraordinary achievements and surprising outcomes [76]. Even in distressing situations or activities, people can achieve positive mental health outcomes [77]. In this direction, high psychological capital can enable young adults living with HIV to approach their developmental tasks better, facilitating the gratification of their BPNs and, consequently, better mental health. In line with the SDT, the study by Datu and colleagues demonstrated that psychological capital is an essential factor in autonomous or intrinsic motivation to pursue career goals [78], hence crucial for realizing the need for competence. Psychological capital is also valuable for exercising self-directed behavior when working in autonomous situations [79] and is necessary for achieving flow experiences in performing tasks [80]. Therefore, psychological capital may enable individuals to focus fully on a given activity or goal and, hence, can exert more effort toward satisfying their BPNs.

By facilitating the attainment of BPNs, psychological capital is likely to result in enhanced mental health of young adults living with HIV. All four psychological resources constituting psychological capital are crucial to interventions seeking to improve wellbeing and psychological health [81]. Studies using psychological capital interventions have revealed substantial improvements in mental health [82–84]. In distressing situations such as during COVID-19, psychological capital was found to be an essential buffer against anxiety and depression and significantly positively affected psychological wellbeing, life satisfaction, and other mental health outcomes [85–87]. Surprisingly, there seems to be limited literature on applying psychological capital to promoting the wellbeing of PWH/AIDS. Available empirical findings suggest that psychological capital plays a vital role in improving mental health in the context of HIV/AIDS [88]. Previous research has focused chiefly on the role of individual facets of psychological capital, especially resilience [89, 90] and hope [91, 92]. However, psychological capital involves more resources that collectively have a more significant impact [33], which has implications for the application of psychological resources in HIV/AIDS interventions.

Based on this literature, we examine the effects of BJW and psychological capital on the satisfaction of BPNs and the mental health of young adults living with HIV through a serial mediation process (See Fig 1). Mental health is analyzed with two dimensions, one representing good mental health (flourishing) and another representing poor mental health (distress), which is in line with some studies establishing the factor structure of the scale [93, 94]. Responses to HIV-positive status are complex, which may also be tagged to how HIV infection was obtained. Blame and anger towards the source (e.g., towards parents in case of perinatal transmission) are common in situations where an individual thinks that he or she did not deserve to get the infection, while self-blame and feelings of personal deservedness are common where individuals feel that their behavior played a role in acquiring the infection [95, 96]. These could lead to lower levels of BJW, psychological capital, developmental need attainment,

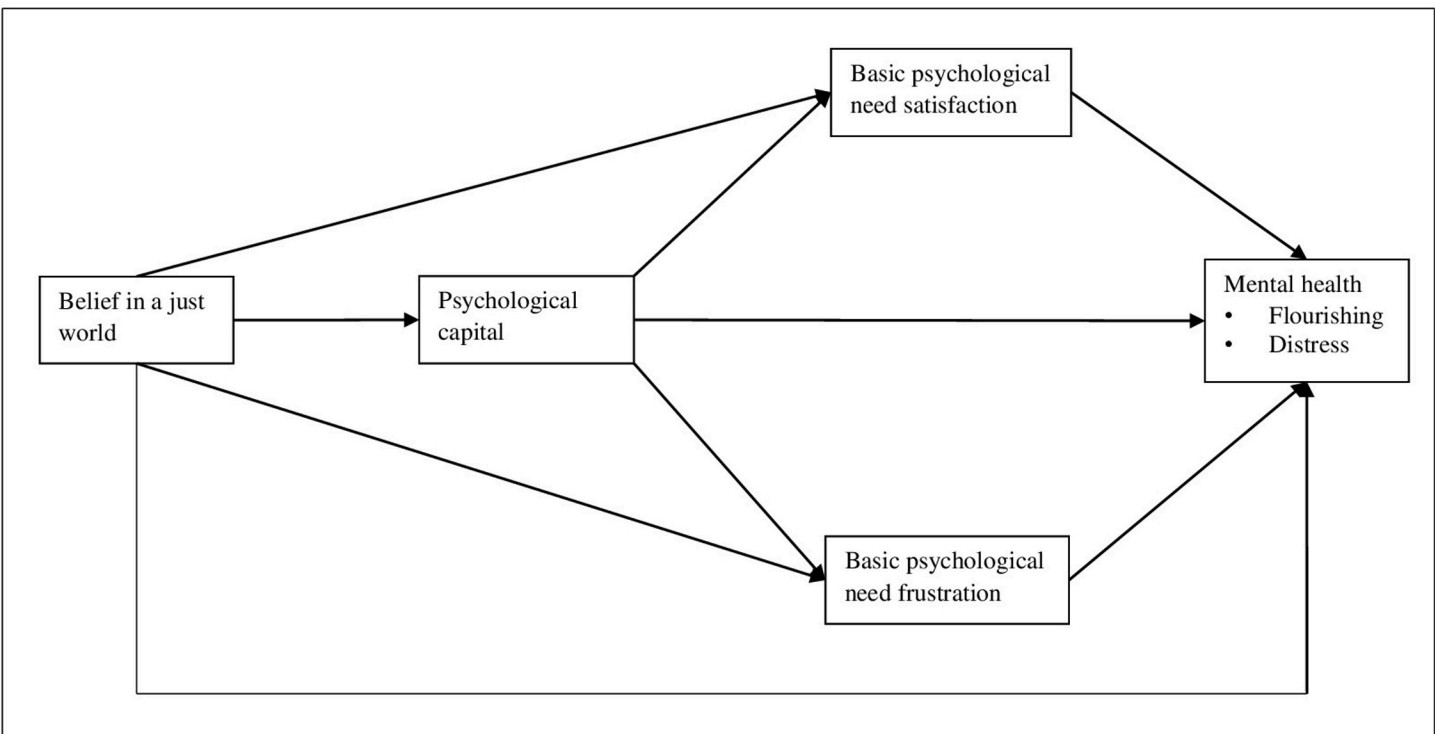

**Fig 1. Presents the hypothesized serial mediation model through which BJW is associated with mental health.**

and mental health among young PWH, and hence, higher chances of BPN frustration and distress.

We, therefore, hypothesized that YALWH will report lower BJW (*H1a*), psychological capital (*H1b*), BPN satisfaction (*H1c*), and mental health -flourishing (*H1d*), as well as higher BPN frustration (*H1e*) and distress (*H1f*) compared to the student sample. Concerning the relationships among the variables, we hypothesized that BJW is positively associated with psychological capital (*H2a*), BPN satisfaction (*H2b*), and good mental health - flourishing (*H2c*) but negatively related to BPN frustration (*H2d*) and poor mental health–distress (*H2e*). Second, psychological capital is also positively associated with BPN satisfaction (*H3a*) and good mental health-flourishing (*H3b*) but negatively related to BPN frustration (*H3c*) and poor mental health–distress (*H3d*). Third, BPN satisfaction is positively associated with good mental health-flourishing (*H4a*), while BPN frustration is positively related to poor mental health-distress (*H4b*). Regarding the mediation process, we hypothesized that the effects of BJW on mental health-flourishing are mediated by psychological capital (*H5a*), BPN satisfaction (*H5b*), or suppressed by BPN frustration (*H5c*). Similarly, the effects of BJW on mental health–distress are mediated by psychological capital (*H6a*), BPN satisfaction (*H6b*), and BPN frustration (*H6c*). We further hypothesized double mediation paths such that the effects of BJW on mental health-flourishing are simultaneously mediated through both psychological capital and BPN satisfaction (*H7a*) or BPN frustration (*H7b*). Similarly, the effects of BJW on mental health-distress are simultaneously mediated through both psychological capital and BPN satisfaction (*H7c*) or BPN frustration (*H7d*). For hypotheses H2 –H7, we tested for differences between the YALWH and student samples.

## Methods

### Participants and procedure

The study compared a sample of YALWH with a sample of students to ascertain the associations among a belief in a just world, psychological capital, developmental outcomes, and mental health. A priori power analysis using G-power [97, 98] for regression analysis with six (6) predictors shows that a minimum of 146 participants is required for each sample category. This sample size is needed at a moderate anticipated effect size of 0.15 [98], a desired error probability level of 0.05, and a desired statistical power of 0.95. Participants were 227 YALWH (81.8% female, 18.2% male) and 187 university students (72.2% female, 27.8% male). Participants included patients attending the HIV/ AIDS clinic at Nurture Africa Medical Centre in Wakiso District, Uganda.

The questionnaire was administered to YALWH with the support of trained clinical officers and nurses during routine clinic visits. The sample of students was obtained from a large university in Uganda. Students were invited to complete an online questionnaire through their mailing lists and WhatsApp groups. The data collection process for this survey study started on August 8, 2023, and ended on December 20, 2023. Overall, 276 students responded to the questionnaire. However, only 187 responses were considered for analysis after screening for incomplete and irresponsible responding [99]. Concerning age, the study targeted only young adults aged 18–39 years from the developmental psychology perspective of young adulthood [5]. The average age of participants was 29.61 years ($SD$ = 6.00) for the YALWH sample and 24.27 years (SD = 4.00) for the student sample.

### Ethical approval and informed consent

This paper reports data from the project titled "Positive Psychological Attributes and Developmental Outcomes of Adults Living with HIV Acquired Vertically or Horizontally in Childhood: Implications for Mental Health, Retention in Care, and Adherence." Ethical approval for this study was obtained from the Makerere University School of Medicine Research and Ethics Committee (Mak-SOMREC-2022-517). All participants were given information about the purpose of the study and the procedure and requested to sign the informed consent forms.

**Measures.** The questionnaire was administered in English; hence, only YALWH that could read and write in English were included in the sample. The measures were administered in their original form with no modifications. A 6-point Likert-type rating scale was used for all measures. Likert-type scales offer greater reliability, and the total scores tend to be more reliable and better reflection of the underlying construct [100, 101].

***Belief in a just world*** was assessed using the 7-item personal BJW scale [28]. A sample item is "I believe that most of the things that happen in my life are fair" (1 = *strongly disagree*, and 6 = *strongly agree*). Previous studies in China and Germany have reported impressive reliabilities ranging from .82 to .88 [50]. The scale revealed acceptable internal consistency for the present study (α = .73 for the YALWH sample and .78 for the student sample).

***Psychological capital*** was assessed using the short 12-item psychological capital questionnaire [102]. The questionnaire measures the four psychological resources that constitute psychological capital: confidence, hope, resilience, and optimism. A sample item is "If I should find myself in a bad situation, I could think of many ways to get out of it" (1 = *strongly disagree*, and 6 = *strongly agree*). The developers report impressive Cronbach's α coefficients ranging from .88 to .89 [103]. They also report impressive discriminant and convergent validity. The questionnaire showed high internal consistency (α = .86 for the YALWH sample and .89 for the student sample).

***Psychological basic needs (PBN) satisfaction and frustration***: were assessed using the basic psychological need satisfaction and frustration questionnaire [104]. Unlike other psychological need measures, this questionnaire measures both the satisfaction and frustration of PBN. The questionnaire consists of 24 items measured on a Likert scale (1 = *strongly disagree*, and 6 = *strongly agree*). Sample items are "I feel a sense of choice and freedom in the things I undertake" for autonomy; "I feel close and connected with other people who are important to me" for relatedness; and "I feel confident that I can do things well" for competence. The questionnaire developers validated it among USA, Peru, China, and Belgium samples. Reliability coefficients were impressive for the USA and Belgium samples but less acceptable for Peru and China [104]. The internal consistency of the questionnaire was impressive for the present study. Concerning psychological need satisfaction, the reliability was α = .81 for the YALWH sample and .90 for the student sample. Reliability for psychological need frustration was α = .80 for the YALWH sample and .86 for the student sample.

***Mental health*** was assessed using the General Health Questionnaire (GHQ-12) to measure mental health [105], which is widely used for screening mental health problems and produces results that are comparable to those obtained from extended versions [106]. The questionnaire comprises 12 items assessing different aspects of mental health in the past six (6) months. In line with studies that have indicated this instrument as a multi-dimensional scale [93, 94, 107], the positive items constituted good mental health (flourishing), while the negative items reflected poor mental health (distress). Sample items are ". . . . . . felt unhappy and depressed" for distress and ". . . . . . felt reasonably happy" for flourishing. All items were rated on a 6-point Likert scale (1 = *strongly disagree*, and 6 = *strongly agree*). The reliability analysis revealed acceptable internal consistency for flourishing (α = .71 for the YALWH sample and .84 for the student sample). Similarly, acceptable internal consistency was observed for the distress dimension (α = .78 for the YALWH sample and .86 for the student sample).

**Data analysis strategy.** The study examines the proposition that a BJW impacts psychological capital and BPN satisfaction and, consequently, mental health differently for YALWH. We first applied a MANOVA analysis to test whether the YALWH and the student sample differed significantly. Regression analysis was used to test the effects of a BJW on psychological capital, BPN satisfaction or frustration, and mental health. Separate regression models were computed for each sample. Regression-based path analysis was performed to test the effects of a BJW on psychological capital, BPN satisfaction, and mental health. Thus, serial mediation regression analyses were conducted in PROCESS Macro model 81 [108], with psychological capital as a first-level mediator and BPN satisfaction and frustration as parallel second-level mediators. Gender and age were added to the regression model as covariates, given that they tend to affect the mental health of PWH [109]. In addition, sample bootstrapping was applied at 5000, as suggested by Hayes [110].

## Results

The descriptive statistics and bivariate correlations for both samples are shown in Table 1. The differences between the two samples in terms of the different study variables are shown in Table 2. Specifically, the sample of YALWH had significantly higher mean scores than the student sample on BJW ($F$ = 74.29, $p$ = .000, $\eta_p^2$ = .153), BPN frustration ($F$ = 7.46, $p$ = .007, $\eta_p^2$ = .018), and distress ($F$ = 9.45, $p$ = .002, $\eta_p^2$ = .022). Hence, hypotheses *H1a* is rejected, while hypotheses *H1e* and *H1f* are supported. There were no substantial differences concerning psychological capital, BPN satisfaction, and mental health (flourishing). Thus, hypotheses *H1b*, *H1c*, and *H1d* are also not supported. Table 3 shows the regression results for the student sample, while Table 4 shows the regression results for the sample of YALWH.

**Table 1. Descriptive statistics and correlations among study variables segregated by samples.**

| | YALWH sample descriptives | | | Student sample descriptives | | | Correlations | | | | | | | |
|---|---|---|---|---|---|---|---|---|---|---|---|---|---|---|
| | *M* | *SD* | α | *M* | *SD* | α | 1 | 2 | 3 | 4 | 5 | 6 | 7 | 8 |
| 1. Gender | _ | _ | _ | | | _ | _ | .35*** | -.19* | -.12 | -.15* | -.04 | -.12 | -.20** |
| 2. Age | 29.61 | 6.00 | _ | 24.27 | 4.00 | _ | .15* | _ | -.004 | .09 | .04 | -.24** | .06 | -.21** |
| 3. Belief in a just world | 4.49 | 0.85 | .73 | 3.81 | 0.75 | .78 | .05 | -.07 | _ | .35*** | .46*** | .11 | .43*** | .14 |
| 4. Psychological capital | 4.60 | 0.85 | .86 | 4.54 | 0.75 | .89 | -.04 | -.14* | .49*** | _ | .74*** | -.25** | .65*** | -.09 |
| 5. PBN satisfaction | 4.72 | 0.74 | .81 | 4.62 | 0.71 | .90 | -.07 | -.15* | .35*** | .60*** | _ | -.30*** | .71*** | -.11 |
| 6. PBN frustration | 3.31 | 1.25 | .80 | 3.00 | 1.01 | .86 | .05 | .27*** | -.06 | -.18** | -.35*** | _ | -.18* | .67*** |
| 7. Mental health (flourishing) | 4.62 | 0.82 | .71 | 4.55 | 0.76 | .84 | -.11 | -.07 | .24*** | .38*** | .60*** | -.24*** | _ | -.11 |
| 8. Mental health (distress) | 3.68 | 1.17 | .78 | 3.33 | 1.13 | .86 | -.05 | .30*** | -.07 | -.19** | -.19** | .61*** | -.02 | _ |

Note:

Correlations for the YALWH sample below the diagonal, correlations for the student sample above the diagonal

*. *p* < .05

**. *p* < .01

***. *p* < .001, N = 414 (total sample), 187 (student), 227 (YALWH)

[a]Male = 0, Female = 1; [b]YALWH = 0, Students = 1

Concerning the direct effects of BJW, we observed significant positive effects in the student sample on psychological capital (*B* = .33, *p* < .001), BPN satisfaction (*B* = .21, *p* < .001), and flourishing aspect of mental health (*B* = .14, *p* < .05). In the YALWH sample, BJW was only substantially related to psychological capital (*B* = .48, *p* < .001). Hence, hypotheses *H2a*, *H2b*, and *H1c* are supported in the student sample. However, only *H2a* is supported in the YALWH

**Table 2. MANOVA for differences between the sample categories.**

| Variables | Sample category | *M* | *SD* | *F* | *p* | $\eta_p^2$ |
|---|---|---|---|---|---|---|
| Belief in a just world | YALWH | 4.49 | 0.85 | 74.29 | .000 | .153 |
| | Students | 3.81 | 0.75 | | | |
| | Total | 4.18 | 0.88 | | | |
| Psychological capital | YALWH | 4.60 | 0.85 | 0.57 | .452 | .001 |
| | Students | 4.54 | 0.75 | | | |
| | Total | 4.57 | 0.81 | | | |
| BPN satisfaction | YALWH | 4.72 | 0.74 | 2.26 | .134 | .005 |
| | Students | 4.62 | 0.71 | | | |
| | Total | 4.68 | 0.73 | | | |
| BPN frustration | YALWH | 3.31 | 1.25 | 7.46 | .007 | .018 |
| | Students | 3.00 | 1.01 | | | |
| | Total | 3.17 | 1.16 | | | |
| MH (flourishing) | YALWH | 4.62 | 0.82 | 0.96 | .331 | .002 |
| | Students | 4.55 | 0.76 | | | |
| | Total | 4.59 | 0.79 | | | |
| MH (distress) | YALWH | 3.68 | 1.17 | 9.45 | .002 | .022 |
| | Students | 3.34 | 1.13 | | | |
| | Total | 3.52 | 1.16 | | | |

Note:

N = 414 (187 students, 227 YALWH), $\eta_p^2$ = Partial Eta squared

**Table 3. Regression results for the student sample.**

| | Psychological capital | | | BPNs satisfaction | | | BPNs frustration | | | Mental health (flourishing) | | | Mental health (distress) | | |
|---|---|---|---|---|---|---|---|---|---|---|---|---|---|---|---|
| | B(SE) | CI | | B(SE) | CI | | B(SE) | CI | | B | CI | | B(SE) | CI | |
| | | LL | UL | | LL | UL | | LL | UL | | LL | UL | | LL | UL |
| Gender[a] | -.18(.12) | -.42 | .07 | -.04(.08) | -.20 | .12 | .11(.17) | -.22 | .44 | -.01(09) | -.19 | .18 | -.42(.15)** | -.71 | -.12 |
| Age | .02(.01) | -.003 | .05 | -.002(.01) | -.02 | .02 | -.06(02)** | -.10 | -.02 | .01(.01) | -.02 | .03 | .003(.02) | -.03 | .04 |
| Belief in a just world (BJW) | .33(.07)*** | .19 | .47 | .21(.05)*** | .12 | .31 | .30(.10)** | .10 | .49 | .14(.06)* | .02 | .26 | .01(.10) | -.18 | .20 |
| Psychological capital (PsyCap) | | | | .62(.05)*** | .53 | .72 | -.40(.10)*** | -.60 | -.21 | .27(.08)*** | .13 | .42 | .04(.12) | -.21 | .28 |
| BPNs satisfaction | | | | | | | | | | .47(.09)*** | .30 | .65 | .07(.14) | -.21 | .34 |
| BPNs frustration | | | | | | | | | | .01(.04) | -.08 | .08 | .77(.07)*** | .63 | .90 |
| *R² statistics* | R² = .14, F = 9.85*** | | | R² = .59, F = 64.83*** | | | R² = .15, F = 8.17*** | | | R² = .55, F = 36.39*** | | | R² = .48, F = 27.37*** | | |
| Total effects | | | | | | | | | | .43(.07)*** | .29 | .56 | .18(.11) | -.04 | .39 |
| Direct effects | | | | | | | | | | .14(.06)* | .02 | .26 | .01(.10) | -.18 | .20 |
| *Standardized indirect effects* | | | | | | | | | | β(Boot SE) | Boot CI | | β(Boot SE) | Boot CI | |
| | | | | | | | | | | | LL | UL | | LL | UL |
| Total indirect effects | | | | | | | | | | .29(.07) | .15 | .41 | .11(.06) | -.01 | .24 |
| Via PsyCap | | | | | | | | | | .09(.03) | .04 | .16 | .01(.02) | -.04 | .05 |
| Via BPNs satisfaction | | | | | | | | | | .10(.04) | .03 | .18 | .01(.02) | -.02 | .04 |
| Via BPNs frustration | | | | | | | | | | .001(.01) | -.03 | .03 | .15(.06) | .05 | .27 |
| Via PsyCap and BPNs satisfaction | | | | | | | | | | .10(.03) | .04 | .16 | .01(.02) | -.02 | .04 |
| Via PsyCap and BPNs frustration | | | | | | | | | | -.001(.01) | -.01 | .01 | -.07(.02) | -.12 | -.03 |

Note:

*. $p < .05$

**. $p < .01$

***. $p < .001$, [a]Male = 1, Female = 2; [b]YALWH = 0, Students = 1; N = 187

CI = 95% Confidence intervals, LL = Lower limit confidence intervals, UL = Upper limit confidence intervals

sample. Whereas we predicted a negative relationship between BJW and BPN frustration (*H2d*), the results show a positive relationship in the student sample (B = .30, p < .01). The direct effects on distress were not significant in both samples; hence, hypotheses *H2e* is not supported.

There were notable differences in the results concerning hypothesis *H3*. In the student sample, psychological capital was positively associated with BPN satisfaction (*B* = .62, *p* < .001) and flourishing (*B* = .27, *p* < .001) and negatively associated with BPN frustration (*B* = -.40, *p* < .001). In the YALWH sample, psychological capital was positively associated with BPN satisfaction (*B* = .48, *p* < .001) and negatively associated with BPN frustration (*B* = -.24, *p* < .05) and distress (*B* = -.19, *p* < .05). Therefore, hypotheses *H3a*, *H3b*, and *H3c* are supported in the student sample. However, *H3d* is not supported in this sample. On the other hand, Hypotheses *H3a*, *H3c*, and *H3d* are supported, while *H3b* is not supported in the YALWH sample.

Concerning the effects of BPNs on mental health, the results in Table 3 showed a positive association between BPN satisfaction and flourishing (*B* = .47, *p* < .001) and a positive association between BPN frustration and distress (*B* = .77, *p* < .001) for the student sample. Similarly, results for the YALWH sample in Table 4 show a positive association between BPN satisfaction and the flourishing aspect of mental health (*B* = 61, *p* < .001) and between BPN frustration and distress (*B* = .55, *p* < .001). The effects of BPN satisfaction on distress and BPN frustration on flourishing were not significant for both samples. Therefore, hypotheses *H4a* and *H4b* are supported.

**Table 4. Regression results for the sample of YALWH.**

| | Psychological capital | | | BPNs satisfaction | | | BPNs frustration | | | Mental health (flourishing) | | | Mental health (distress) | | |
|---|---|---|---|---|---|---|---|---|---|---|---|---|---|---|---|
| | B(SE) | CI | | B(SE) | CI | | B(SE) | CI | | B | CI | | B(SE) | CI | |
| | | LL | UL | | LL | UL | | LL | UL | | LL | UL | | LL | UL |
| Gender[a] | -.10(.12) | -.34 | .14 | -.07(.10) | -.27 | .12 | -.01(.20) | -.38 | .40 | -.16(11) | -.37 | .06 | -.30(.15) | -.60 | -.01 |
| Age | -.01(.01) | -.03 | .002 | -.01(.01) | -.02 | .01 | .05(.01)*** | .03 | .08 | .01(01) | -.01 | .02 | .03(.01)** | .01 | .05 |
| Belief in a just world (BJW) | .48(.0)*** | .37 | .60 | .07(.05) | -.04 | .17 | .06(.11) | -.16 | .27 | .04(.06) | -.08 | .16 | .01(.08) | -.15 | .17 |
| Psychological capital (PsyCap) | | | | .48(.05)*** | .37 | .59 | -.24(.11)* | -.46 | -.03 | .03(.07) | -.11 | .16 | -.19(.10)* | -.38 | -.003 |
| BPNs satisfaction | | | | | | | | | | .61(.08)*** | .46 | .77 | .17(.11) | -.04 | .38 |
| BPNs frustration | | | | | | | | | | -.03(.04) | -.11 | .05 | .55(.05)*** | .44 | .65 |
| *R² statistics* | $R^2 = .25$, $F = 25.09$*** | | | $R^2 = .37$, $F = 32.89$*** | | | $R^2 = .10$, $F = 6.00$*** | | | $R^2 = .37$, $F = 21.81$*** | | | $R^2 = .41$, $F = 25.90$*** | | |
| Total effects | | | | | | | | | | .24(.06)*** | .11 | .36 | -.06(.09) | -.23 | .11 |
| Direct effects | | | | | | | | | | .04(.06) | -.08 | .16 | .01(.08) | -.15 | .17 |
| *Standardized indirect effects of BJW* | | | | | | | | | | β(Boot SE) | Boot CI | | β(Boot SE) | Boot CI | |
| | | | | | | | | | | | LL | UL | | LL | UL |
| Total indirect effects | | | | | | | | | | .21(.05) | .12 | .30 | -.06(.06) | -.16 | .06 |
| Via PsyCap | | | | | | | | | | .01(.03) | -.05 | .07 | -.07(.03) | -.13 | -.01 |
| Via BPNs satisfaction | | | | | | | | | | .04(.03) | -.01 | .10 | .01(.01) | -.01 | .03 |
| Via BPNs frustration | | | | | | | | | | -.002(.01) | -.02 | .01 | .02(.05) | -.07 | .12 |
| Via PsyCap and BPNs satisfaction | | | | | | | | | | .15(.03) | .10 | .21 | .03(.02) | -.002 | .07 |
| Via PsyCap and BPNs frustration | | | | | | | | | | .004(.01) | -.01 | .02 | -.05(.02) | -.09 | -.01 |

Note:

*. $p < .05$

**. $p < .01$

***. $p < .001$, [a]Male = 1, Female = 2; [b]YALWH = 0, Students = 1; N = 227

CI = 95% Confidence intervals, LL = Lower limit confidence intervals, UL = Upper limit confidence intervals

In hypotheses 5–7, we propose that the effects of BJW on mental health are mediated by psychological capital and BPN satisfaction or frustration. There are differences between the student sample and the sample of YALWH regarding the mediating effects of psychological capital in the relationship between BJW and mental health. Psychological capital mediated the effects of BJW on flourishing ($\beta$ = .09, *Boot 95% CI* [.04, .16]) but not distress in the student sample. On the other hand, psychological capital mediated the effects of BJW on distress ($\beta$ = -.07, *Boot 95% CI* [-.13, -.01]) but did not mediate the effects on flourishing in the YALWH sample. Therefore, hypothesis H5a is supported for the student sample but not for the YALWH sample. On the contrary, hypothesis H6a is supported in the YALWH sample but not in the student sample.

There were also notable differences between the student and YALWH samples regarding the mediating effects of BPN satisfaction or frustration in the relationship between BJW and mental health. In the student sample, BPN satisfaction mediated the effects of BJW on flourishing ($\beta$ = .10, *Boot 95% CI* [.03, .18]) but not on distress. On the contrary, BPN satisfaction did not mediate the effects of BJW on both the flourishing and distress dimensions of mental health in the YALWH sample. Hence, hypothesis H5b is supported in the student sample, not the YALWH sample. On the contrary, *H6b* is supported in the YALWH sample and not in the student sample. Concerning hypotheses *H5c* and *H6c*, results in Table 3 show that BPN frustration did not mediate the effects of BJW on the flourishing dimension of mental health but mediated the effects on distress in the student sample ($\beta$ = .15, *Boot 95% CI* [.05, .27]). On the

other hand, BPN frustration did not mediate the effects on either flourishing or distress in the YALWH sample (Table 4).

Hypothesis H7 proposes a seria mediation path through which BJW is associated with mental health. Results show that a double mediation path of the effects of BJW via psychological capital and BPN satisfaction was significant for flourishing in both samples: the student sample ($\beta$ = .10, *Boot 95% CI* [.04, .16]) and the YALWH sample ($\beta$ = .15, *Boot 95% CI* [.10, .21]). These results support hypothesis *H7a*. However, *H7c* is not supported. Concerning the second double mediation path, we observe significant effects of BJW via psychological capital and BPN frustration on distress for both the student sample ($\beta$ = -.07, *Boot 95% CI* [-.12, -.03]) and the YALWH sample ($\beta$ = -.05, *Boot 95% CI* [-.09, -.01]). However, this double mediation path is not significant for flourishing. Hence, hypothesis *H7b* is not supported, while *H7d* is supported.

## Discussion

The study examined how BJW, psychological capital, and BPN satisfaction or frustration relate to the mental health of YALWH in Uganda, comparing them with a sample of university students. We analyzed a serial mediation model with psychological capital and BPN satisfaction or frustration as possible mediating links between BJW and mental health. The results revealed significant differences between YALWH and students concerning scores on BJW, BPN frustration, and distress. Whereas YALWH reported higher levels of BJW, they also reported higher levels of BPN frustration and distress, representing poor mental health.

These results indicate that BJW and how it relates to developmental and mental health outcomes tend to be complex in the context of HIV. It is possible to have a strong BJW and yet have poor mental health since BJW is associated with both positive and negative aspects of wellbeing [111]. Previous research on BJW has focused on how other people react to individuals living with HIV. Specifically, the understanding is that BJW is mainly correlated with attitudes such as isolationist, lifestyle change, and protectionist, all of which suggest discrimination and stigmatization towards PWH [112]. However, the concepts of self-blame, shame, and personal deservedness are popular in understanding the mental health of PWH [58, 96, 113]. These not only reflect self-stigma. Hence, BJW can promote psychological functioning and mental health if one deploys its adaptive functions or result in psychological dysfunction if BJW facilitates negative reactions towards self, such as self-blame.

Generally, the two positive psychological attributes of BJW and psychological capital were significantly associated with satisfaction of BPNs and mental health. However, the results show an interesting pattern of associations. First, as expected, psychological capital was positively associated with BPN satisfaction and negatively with BPN frustration in both samples. However, it was only positively related to mental health (flourishing) in the student sample but not in the YALWH sample, suggesting that the effects of psychological capital on the mental health of YALWH could be fully mediated by BPN satisfaction or frustration. However, psychological capital is negatively associated with distress in this sample. These results reaffirm the role of psychological capital in enhancing mental health and wellbeing [81, 86, 114, 115]. The psychological resources that make up psychological capital are essential for pursuing and attaining goals [75, 76], which translates into improved mental health. Moreover, this seems to be more pronounced among the YALWH. Psychological capital involves protective resources such as resilience, which buffers against the impacts of HIV stigma [116, 117], improving psychological functioning and health. In this way, psychological capital could also be essential for remaining in HIV care services and adherence to antiretroviral treatment.

The most interesting patterns, however, relate to the direct effects of BJW. Whereas it is positively associated with desirable outcomes, including satisfaction of BPNs and good mental health (flourishing), it is also positively associated with undesirable outcomes, especially BPN frustration. BJW is considered a personal adaptive resource [118], thus essential for psychological health. The adaptive qualities involved in BJW can enable PWH to seek satisfaction of their psychological needs and, consequently, improve their mental health. Although the effects of BJW on adverse outcomes such as BPN frustration and distress are not substantial, the likely dark side of BJW should not be ignored. A strong BJW can result in internalizing problems. For example, self-blame, personal deservedness, and other internal feelings

reflect self-faulting among PWH [96, 113]. In this regard, people can perceive their HIV-positive status and related challenges as deserved punishments for their behaviors, in turn exacerbating malfunctioning and mental health problems. Previous research has chiefly associated personal BJW with blaming others [119] and blaming the victim [120, 121]. On the other hand, personal BJW can lead to the use of self-blame as a coping strategy, facilitating self-stigmatization.

An important contribution of this study is the differentiation in how BPN satisfaction or frustration affects mental health. Our results show that in both samples, satisfaction of BPNs is positively associated with the flourishing aspect of mental health but not related to the distress aspect. Conversely, BPN frustration is positively associated with the distress aspect but not associated with the flourishing aspect. In line with the SDT [14, 39], these findings highlight the role of satisfying BPNs in mental health. Whereas satisfying BPNs improves mental health, frustrating BPNs is likely to result in psychological distress. Satisfying BPNs facilitates psychological functioning and growth [13, 14]. On the other hand, BPN frustration results in maladjustment, which is associated with costs, including disengagement, poor wellbeing, and distress [61]. Moreover, the effects of satisfying or frustrating BPNs on mental health were quite similar in both student and YALWH samples, suggesting that the role of satisfying BPNs on mental health does not depend on HIV status, reinforcing the idea that BPNs are universal and everyone strives to satisfy them [11, 14, 61].

The pattern above, in the effects of BPN satisfaction and frustration, is also reflected in the pattern of the serial mediation paths for both samples. Specifically, the effects of BJW on the flourishing aspect of mental health were serially mediated by psychological capital and BPN satisfaction. On the other hand, the effects of BJW on the distress dimension were serially mediated by psychological capital and BPN frustration. BPN need satisfaction is considered a protecting factor for mental health [122], which may be activated by the adaptive aspects of BJW and the mental strength that characterizes psychological capital. Therefore, whereas BPN satisfaction may transmit the positive effects of BJW and psychological capital to sustain or improve mental health, BPN frustration is likely to activate the dark aspect of BJW that causes distress and may suppress the expected protective effects of psychological capital on mental health.

The study findings have several practical implications. First, whereas BJW is generally considered to espouse adaptive qualities that enable coping and exercising resilience [31, 46, 47] and, therefore, essential for mental health, our findings indicate that there is a dark side to the belief that the world is and fair and thus people deserve what they get. Whereas BJW may enable people to feel good about themselves, hence improving mental health, on the other hand, it can facilitate PWH to consider the adverse events happening to them as deserved, thus increasing risks for distress. Second, the results show that satisfying BPNs is associated with flourishing mental health. On the other hand, BPN frustration is associated with distress. Therefore, in designing BJW interventions, attention should be paid to possible unintended consequences such as self-blame. In addition, mental health programs for YALWH must

incorporate a focus on supporting them to satisfy their BPNs. According to the SDT [39], satisfying these needs fosters psychological growth, thus essential for several life outcomes, including happiness, life satisfaction, wellbeing, and mental health [61, 123, 124]. Third, similar to other studies, our results show that psychological capital is essential in satisfying BPNs and mental health across populations. Psychological capital interventions [125] can be adopted in mental health and care interventions for PWH. Fourth, given the contradicting effects of BJW on BPN satisfaction or frustration and mental health, it might be essential for mental health practitioners, especially those working with PWH, to facilitate clients to cherish positive and adaptive justice beliefs relating to their situations and identify and expunge the maladaptive justice beliefs about themselves.

## Limitations and directions for future studies

Despite the above theoretical and practical implications of the study, our findings should be applied in the context of certain limitations. First, the hypotheses are tested with cross-sectional data. Therefore, causal conclusions cannot be drawn with certainty. In addition, challenges relating to survey questionnaires using self-report measures, such as common methods bias [126, 127], cannot be ruled out.

Second, the study mainly focuses on YALWH and uses the student sample as a comparison group. Although both samples fall within the same developmental stage, many participants in the YALWH sample had lower levels of education and were, on average, older than the students. In addition, we did not collect information relating to the educational attainment and occupational status of the participants in the YALWH sample. Psychological constructs such as BJW are sensitive to socio-demographic variables. Therefore, future studies could benefit from using a comparable sample within the same community settings and comparing constructs such as BJW, BPNs, and mental health against the differences in socio-demographic variables between the student and YALWH samples. In addition, although only 1.6% of participants in the student sample were HIV positive, we did not control for the HIV status of this sample, which could contribute to the insignificant difference between the two samples on some of the study variables, including psychological capital, BPN satisfaction, and mental health–flourishing dimension.

Third, positive psychology attributes such as psychological capital are state- rather than trait-like and can be affected by events. Therefore, longitudinal studies would be best to test the impact of such attributes. Future studies could also benefit from using intervention research approaches to identify the long-term effects of such variables and their application in psychosocial intervention programs for PWH. Fourth, the results suggest that BJW could have a negative side concerning how it is associated with BPNs and mental health outcomes. Confirmatory and qualitative studies are required to generate detailed explanations of how BJW relates to both positive and negative developmental and mental health outcomes.

## Conclusion

Young adulthood is a developmental stage in which people seek to achieve some of life's most important tasks, such as establishing a career, becoming autonomous, and establishing meaningful and intimate relationships, representing the BPNs. In line with the SDT theory, failure to fulfill BPNs can result in mental health problems. Our results demonstrate that positive psychological attributes, especially BJW and psychological capital, could be valuable resources for satisfying BPNs and are important to mental health. From the findings of this study, the role of BJW and psychological capital in satisfying BPNs and mental health could apply across populations, including YALWH and students. Hence, interventions that strengthen these positive

attributes must be incorporated into psychosocial programs, especially for PWH. However, our results suggest that care must be observed in actions toward boosting BJW because it tends to correlate positively with both desired and undesired developmental and mental health outcomes.

## Supporting information

**S1 Data.**
(RAR)

## Acknowledgments

The authors thank Nurture Africa Medical Centre management for allowing us to conduct the study at their facility and the medical officers and nurses who supported the data collection process. We also thank YALWH and the students who participated in the study.

## Author Contributions

**Conceptualization:** Martin Mabunda Baluku, Samuel Ouma, John Kiweewa.

**Data curation:** Martin Mabunda Baluku, Samuel Ouma, Gerald Mukisa Nsereko.

**Formal analysis:** Martin Mabunda Baluku, John Kiweewa.

**Funding acquisition:** Martin Mabunda Baluku, John Kiweewa.

**Investigation:** Martin Mabunda Baluku, Gerald Mukisa Nsereko, Stuart Kwikiriza.

**Methodology:** Martin Mabunda Baluku, Samuel Ouma, Joanita Nangendo.

**Project administration:** Martin Mabunda Baluku, Brian Iredale, Gerald Mukisa Nsereko, Joanita Nangendo.

**Resources:** Martin Mabunda Baluku, Brian Iredale, Joanita Nangendo, John Kiweewa.

**Supervision:** Samuel Ouma, Stuart Kwikiriza, John Kiweewa.

**Validation:** Joanita Nangendo, Stuart Kwikiriza.

**Writing – original draft:** Martin Mabunda Baluku.

**Writing – review & editing:** Martin Mabunda Baluku, Samuel Ouma, Brian Iredale, Gerald Mukisa Nsereko, Joanita Nangendo, Stuart Kwikiriza, John Kiweewa.

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
