## [Decision Letter · Decision Letter 0]

11 Jun 2024

PMEN-D-24-00143

Linking Belief in a Just World and Psychological Capital to Developmental Tasks Attainment and Mental Health of Young Adults Living with HIV: A Comparative Analysis

PLOS Mental Health

Dear Dr. Baluku,

Thank you for submitting your manuscript to PLOS Mental Health. After careful consideration, we feel that it has merit but does not fully meet PLOS Mental Health’s publication criteria as it currently stands. Therefore, we invite you to submit a revised version of the manuscript that addresses the points raised during the review process.

We look forward to receiving your revised manuscript.

Kind regards,

Kaaren R Mathias, PhD

Academic Editor

PLOS Mental Health

Journal Requirements:

https://journals.plos.org/mentalhealth/s/figures 

https://journals.plos.org/mentalhealth/s/figures#loc-file-requirements 

Additional Editor Comments (if provided):

Dear Authors,

Thank you for your generally well written article and thoughtful study. We have suggested that we could re-consider this paper after you make major revisions which are outlined by Reviewers 1 and 2. I agree with both reviewers that you have somewhat lost the wood for the trees and greater clarity required in results and discussion while avoiding extraneous detail. In particular, look at writing in ways that are clear and succinct. For example - these two sentences in the conclusion are somewhat repetitive in concept and you could give a hard edit throughout the document to use a more active tense and journalistic brisk style. Please edit the whole manuscript with this in mind.

In line with the SDT theory, failure to fulfill these tasks can result in mental health problems. The challenge is even greater for YALWH given that they have extra mental health challenges that they are confronted with.

Please review your Discussion and in particular strengthen the implications section by discussion implications for clinicians and public mental health practitioners or others.

Additionally please address all the comments from Reviewers 1 and 2, point by point.

Thank you,

Dr Kaaren Mathias (Handling editor)

Reviewers' comments:

Reviewer's Responses to Questions

**Comments to the Author**

1. Does this manuscript meet PLOS Mental Health’s publication criteria? Is the manuscript technically sound, and do the data support the conclusions? The manuscript must describe methodologically and ethically rigorous research with conclusions that are appropriately drawn based on the data presented.

Reviewer #1: Partly

Reviewer #2: Partly

2. Has the statistical analysis been performed appropriately and rigorously?

Reviewer #1: Yes

Reviewer #2: Yes

3. Have the authors made all data underlying the findings in their manuscript fully available (please refer to the Data Availability Statement at the start of the manuscript PDF file)?

Reviewer #1: No

Reviewer #2: No

4. Is the manuscript presented in an intelligible fashion and written in standard English?

Reviewer #1: Yes

Reviewer #2: Yes

5. Review Comments to the Author

Reviewer #1: The manuscript chief drawback is the inadequate addressing of potential confounder, additionally I would recommend the following correction/improvement

1. Streamline the Introduction: The current introduction is extensive and repetitive. Please condense this section to succinctly cover the scientific background and rationale. Articulate the study's specific objectives and hypotheses clearly and concisely, focusing solely on content that directly supports the study's goals.

2. Clarification of Key Terms: Define critical terms such as "young adult" in the abstract or title to remove ambiguity. Also, specify the age range considered for young adults early in the introduction to provide a clear demographic context.

3. Address Misconceptions Around BJW: It is vital to discuss potential negative perceptions of BJW among people with HIV, particularly concerning self-blame and guilt. Clarify why BJW should not be construed negatively and its implications on self-perception among HIV-positive individuals.

4. Review and Justify Sample Size Calculations: Reevaluate the sample size requirements based on the specified effect size, power, and number of predictors. Justify any deviations from typical sample size requirements or address potential limitations if the current sample is underpowered. Discrepancies in the sample size calculations as per the parameters provided were noted; please rectify or clarify these figures.

5. Enhance Comparative Analysis: Introduce a table detailing the baseline characteristics of both the YALWH and control groups (university students) to ensure comparability. Include variables such as age, occupation, marital status, and socioeconomic status, which are crucial for identifying potential confounders.

6. Specify and Elaborate on Regression Models: Ensure that your regression models are correctly specified. Justify the choice and treatment of all covariates, considering potential confounders like socioeconomic status or duration of HIV diagnosis. Provide a detailed explanation or a step-by-step breakdown of complex statistical methods like serial mediation analysis, possibly in supplementary materials, to enhance

replicability.

7. Explore Interaction Effects: Consider adding interaction terms in your analyses to examine how BJW impacts may vary by demographic factors such as age and educational level within your sample groups.

8. Rationalize Use of Likert Scales: Provide a rationale for treating Likert scale responses as interval data. Discuss the methodological implications of this decision and ensure alignment with current psychological research practices.

9. Clarify Gender Coding: Explain the rationale behind coding gender as a numerical variable and discuss how this approach affects your analysis, ensuring methodological soundness.

10.Improve Table Clarity and Relevance: Enhance the clarity and utility of all tables in the manuscript. Each table should clearly support and justify its inclusion through direct relevance to the study's findings.

11. Justify Control Group Selection: Provide detailed reasoning for using university students as a control group. Discuss their comparability to the YALWH group in terms of educational, geographic, and potentially religious or spiritual affiliations.

12. Expand Theoretical Context in Discussion: Broaden the discussion section to link your findings explicitly with wider psychological theories beyond BJW and psychological capital, such as theories of stigma and identity, to deepen the interpretation of how BJW impacts mental health among YALWH.

13. Functional Data Link: The link provided for accessing the data is nonfunctional. Please provide a working link to ensure accessibility and transparency of your data.

Reviewer #2: Thank you for the opportunity of reviewing the manuscript “Linking Belief in a Just World and Psychological Capital to Developmental Tasks Attainment and Mental Health of Young Adults Living with HIV: A Comparative Analysis” (Reference: PMEN-D-24-00143), submitted to PLOS Mental Health.

Understanding the dimensions and processes that promotes resilience and positive adaptation in individuals with chronic conditions (such as it is the case of HIV) is naturally of great relevance, for public health, social cohesion and mental health.

Therefore, I consider that this manuscript addresses a relevant question. However, I think that the contribution of the manuscript for the field is jeopardized by several shortcomings. In fact, the understanding of the meaning of the results requires additional clarification.

First, and very importantly, authors are studying several constructs (belief in a just world, psychological capital, mental distress, etc.) which we know that are influenced by individual socio-demographic characteristics. And about this, we only know that the group of students had an age average of 24.74 lower than the group of YALWH (of 30.35, SD=5.98), that the student group were university students and that the participants were recruited from routine HIV/AIDS clinic visits. What is the educational status of the group of YALWH? What is their occupational status (probably, most of them have jobs; if yes, what type of occupation do they have)? Because of the importance of individual socio-demographic variables for the variables under study, a detailed description of the 2 groups of participants if fundamental.

Second, the Measures section is insufficient to understand the meaning of the results. No information is described about the language of the instruments: were the items in English? If yes, how was the fluency of the participants in the language? Besides, how was the process of adaptation of the instruments to the Uganda population? No examples of items are provided, so what they are measuring is not clear. Curiously, all the measures performed worse in the YALWH than in the students group. For example, in the mental health scales, α = .75 for the total sample, .84 for the student sample, and .65 for the sample; meaning a marked difference of .20 in reliability between the two groups. Naturally, this raises important concerns about the validity of the results.

Particularly worrying is the case of the variable of “Developmental outcomes”. Authors describe “the attainment of the three developmental tasks was measured using the basic psychological need satisfaction and need frustration questionnaire”. Developmental outcomes are not equivalent to Basic Psychological Needs Satisfaction and Need Frustration. This questionnaire captures individuals Satisfaction and Frustration with 3 basic psychological needs of Relatedness, Autonomy and Competence. Although authors describe the Self-determination Theory in the Introduction, the equivalence that the authors make between Developmental outcomes, “the attainment of three developmental tasks” and Satisfaction/ Frustration with basic psychological needs is misleading. Again, authors make “developmental outcomes”, “attainment of developmental tasks” and “Safisfaction/ Frustration with basic psychological needs”, equivalent, which is not accurate and is misleading.

Third, the understanding of the relations amongst the variables in the 2 groups would benefit from an integration into a single model. For example, did the authors considered running a mediation moderated model (in which the variable “group” is included as moderator of the relationships amongst the variables?

Finally, the authors identify the variable of “Psychological Wellbeing” as the outcome variable. However, in the measures section, there is no reference to the variable of “Psychological wellbeing”. I imagine that that authors are estimating the variable “Psychological Wellbeing” using the two indicators of Mental Health. However, no information is presented about this in the Methods. This is not a minor issue. In fact, I worry that authors use concepts as if they were equivalent (in their meaning, operationalization, and measurement). The manuscript lacks consistency in what concerns the use of concepts and constructs, and this requires particular attention and revision.

6. PLOS authors have the option to publish the peer review history of their article (what does this mean?). If published, this will include your full peer review and any attached files.

**Do you want your identity to be public for this peer review?** For information about this choice, including consent withdrawal, please see our Privacy Policy.

Reviewer #1: **Yes: **abid rizvi

Reviewer #2: **Yes: **Paulo A. S. Moreira

---

## [Editor Report · Decision Letter 1]

15 Aug 2024

Linking Belief in a Just World and Psychological Capital to Psychological Basic Needs Satisfaction and Mental Health of Young Adults Living with HIV:

A Comparative Analysis

PMEN-D-24-00143R1

Dear Dr. Baluku,

We are pleased to inform you that your manuscript 'Linking Belief in a Just World and Psychological Capital to Psychological Basic Needs Satisfaction and Mental Health of Young Adults Living with HIV:

A Comparative Analysis' has been provisionally accepted for publication in PLOS Mental Health.

Best regards,

Kaaren R Mathias, PhD

Academic Editor

PLOS Mental Health

Thank you for your thoughtful responses to the reviewer comments, we are reviewing the word length guidelines and statistics / tables in greater detail and will revert soon about the final status of this article.

Thank you,

Kaaren Mathias